# Nanomedicine-Driven Therapeutic Strategies for Rheumatoid Arthritis-Associated Depression: Mechanisms and Pharmacological Progress

**DOI:** 10.3390/ph19010094

**Published:** 2026-01-04

**Authors:** Jiaxiang Hu, Mingqin Shi, Miao Tian, Baiqing Xie, Yi Tan, Dongxu Zhou, Tengfei Qian, Dongdong Qin

**Affiliations:** 1Jinghua Academy of Zhejiang Chinese Medicine University, Jinghua 321015, China; hujiaxiangwork@163.com; 2Key Laboratory of Traditional Chinese Medicine for Prevention and Treatment of Neuropsychiatric Diseases, Yunnan University of Chinese Medicine, Kunming 650500, China; shimingqin1998@163.com (M.S.); 18332791312@163.com (M.T.); rhinelort@163.com (B.X.); 13452729933@163.com (Y.T.); zdx2432924237@163.com (D.Z.); 3The People’s Hospital of Mengzi, The Affiliated Hospital of Yunnan University of Chinese Medicine, Mengzi 661100, China

**Keywords:** rheumatoid arthritis, depression, nanoparticles, novel drug delivery systems, therapeutic efficacy

## Abstract

Rheumatoid arthritis (RA) is frequently accompanied by depression, a comorbidity arising from the interplay of chronic systemic inflammation, neuroimmune activation, oxidative stress, and dysregulation of the gut–brain axis. Increasing evidence suggests that nanomedicine offers unique opportunities for the integrated management of RA-associated depression by enabling precise modulation of both peripheral inflammation and central nervous system (CNS) pathology. This review outlines the biological mechanisms linking RA and depression—including cytokine cascades, mitochondrial dysfunction, reactive oxygen species (ROS) accumulation, and microbial metabolite imbalance—and highlights recent progress in nanocarrier platforms capable of dual-site intervention. Liposomes, polymeric nanoparticles (NPs), exosomes, inorganic nanozymes, and emerging carbon-based nanomaterials have demonstrated the ability to target inflamed synovium, reprogram macrophage phenotypes, traverse the blood–brain barrier (BBB), suppress microglial overactivation, enhance neuroplasticity, and restore gut microbial homeostasis. Furthermore, stimulus-responsive nanoplatforms activated by ROS, pH, enzymes, or hypoxia provide spatiotemporally controlled drug release, thereby improving therapeutic precision. Finally, we discuss integrative designs such as dual-targeting nanomedicines, co-delivery systems, and microbiota-modulating nano-interventions, which offer promising strategies for the comprehensive treatment of RA-associated depression. This review aims to provide mechanistic insights and design principles to guide the development of next-generation nanomedicine for coordinated systemic-central modulation in RA comorbidity.

## 1. Introduction

Depression is among the most prevalent psychiatric comorbidities in RA, with evidence indicating that up to 50% of RA patients may develop some form of depressive disorder. Clinically, depression exhibits significant heterogeneity, ranging from mild to severe, with major depressive disorder (MDD) representing one of the most debilitating phenotypes [1]. Key clinical features include apathy and persistent negative affective states such as sustained sadness and hopelessness. Although existing therapeutic interventions—including psychotherapy and antidepressant pharmacotherapy—offer substantial benefits, numerous challenges persist. Treatment resistance frequently develops, and relapse following therapy discontinuation is common. Furthermore, given the intricate pathophysiology of depression involving multiple biological systems, the identification of optimal individualized treatment regimens remains complex [2].

Based on the aforementioned background, RA-associated depression is intricately intertwined with clinical manifestations and pathogenic mechanisms, posing complex challenges for treatment that span across systems and targets. Currently, in clinical practice, treatments for RA and depression are predominantly employed independently, a compartmentalized intervention approach that increasingly reveals limitations in efficacy, safety, and long-term management. Standard therapies for RA include traditional disease-modifying anti-rheumatic drugs (DMARDs) such as methotrexate (MTX), biologic agents, and corticosteroids, with long-term systemic administration often carrying risks of hepatotoxicity, nephrotoxicity, immunosuppression, and drug resistance [3,4,5]. Pharmacotherapy for depression primarily relies on selective serotonin (5-HT) reuptake inhibitors (SSRIs) and other antidepressants, which are often characterized by slow onset, notable individual variability in response, and insufficient efficacy against inflammation-driven depression [6,7]. More critically, many antidepressants face significant limitations in BBB permeability, resulting in inadequate CNS drug concentrations [8].

NPs have emerged as versatile drug delivery vectors capable of transporting various therapeutics into targeted cells. Advances in nanotechnology have expanded prospects for both diagnostic and therapeutic applications in biomedicine. Inorganic NPs, in particular, demonstrate promising potential in drug targeting, controlled release systems, and in enhancing the solubility and bioavailability of poorly soluble pharmacological agents. Certain NPs also possess intrinsic anti-inflammatory and antioxidant properties, with demonstrated efficacy across a spectrum of chronic diseases. Concurrently, the progress in nanomedicine introduces innovative technological strategies to surpass the constraints of traditional therapeutic modalities. Nanocarrier platforms improve drug solubility, stability, and pharmacokinetic parameters, significantly elevating targeted tissue drug accumulation while reducing systemic toxicity. Importantly, these nanocarriers can utilize stimuli-responsive mechanisms-such as pH, temperature, or enzyme triggers-to facilitate precise drug delivery within inflammatory microenvironments and the CNS, enhancing therapeutic efficacy and specificity [9]. This review aims to systematically elucidate the key pathological mechanisms underlying RA-associated depression, with a focus on the roles of inflammatory responses, oxidative stress, mitochondrial dysfunction, and gut microbiota imbalance in the development and progression of comorbidity. Building on this, it comprehensively summarizes recent advances in organic nanocarriers, inorganic nanomaterials, exosomes, and biomimetic nanosystems in the treatment of RA and depression, emphasizing their design principles and potential advantages for dual joint-central targeting. By integrating mechanistic insights with nanomedicine design strategies, this work provides a novel theoretical framework and strategic reference for systematic interventions targeting RA-associated depression.

## 2. Pathophysiological Mechanisms of RA-Associated Depression

Although the underlying pathophysiological pathways connecting RA and depression are not fully characterized, key contributors such as systemic inflammation, oxidative stress, and dysbiosis of the gut microbiota are extensively recognized. Both pathologies exhibit an amplified systemic inflammatory state and alterations in pro-inflammatory cytokine networks (Figure 1). Evidence suggests that mitigating oxidative activation in RA could potentially reduce the incidence of depression related to RA [10].

### 2.1. Inflammatory Response

RA is a chronic systemic autoimmune disease characterized by persistent synovial inflammation and immune system hyperactivation. In RA patients, cytokines such as interleukin-6 (IL-6), tumor necrosis factor-alpha (TNF-α), and interleukin-1 beta (IL-1β) are significantly upregulated in synovial tissue and peripheral circulation [11]. These cytokines facilitate synoviocyte proliferation, joint destruction, and propagate downstream immune cascades through multiple mechanistic pathways. Elevated levels of these pro-inflammatory mediators are also observed in the peripheral blood and cerebrospinal fluid of individuals with depression, indicating their potential role in central neuroinflammation and neuroimmune dysregulation within the CNS [12,13]. Peripheral cytokines act as a critical interface between RA and depression, gaining CNS access via humoral and neural routes. Cytokines can traverse the BBB at regions with natural permeability, such as circumventricular organs, or through BBB compromise induced by inflammation, which promotes transendothelial migration of immune cells and cytokines [14,15]. For example, TNF-α disrupts endothelial tight junction integrity by activating soluble guanylate cyclase (sGC) and protein tyrosine kinase (PTK), thereby increasing BBB permeability [16]. Via the neural pathway, afferent sensory nerve endings encircling the inflamed synovial tissue exhibit cytokine receptor expression, facilitating the transmission of peripheral inflammatory signals through the vagus nerve to central autonomic structures such as the nucleus tractus solitarius and the hypothalamus [13]. Neurons within the dorsal root ganglia (DRG) express interleukin-1 receptor (IL-1R), IL-6R, and tumor necrosis factor (TNF) receptors at their peripheral terminals, enabling direct detection of inflammatory mediators [1]. Once within the CNS, cytokines stimulate microglia—the primary producers of central pro-inflammatory mediators—which then secrete elevated levels of inflammatory cytokines and mediators [17]. Microglia-derived IL-1 and TNF-α further activate astrocytes [18]. Persistent chronic inflammation continuously activates the hypothalamic–pituitary–adrenal (HPA) axis and sympathetic nervous system, thereby elevating the risk of depressive disorders [19]. In RA, prolonged inflammatory stress elevates levels of CRH and cortisol [12]. Concurrently, calcium/calmodulin-dependent protein kinase II (CaMKII) activated by excitotoxic stress in the hippocampus enhances cyclooxygenase-2 (COX-2) expression and prostaglandin E2 synthesis, thereby intensifying synovial inflammatory responses [20].

Chronic systemic inflammation in RA facilitates central neuroinflammation through increased peripheral cytokine levels, modulation of BBB permeability, and neuroimmune signaling cascades. These inflammation-induced modifications in neurochemical balance and synaptic plasticity constitute a fundamental mechanistic framework for RA-associated depression. Notably, inflammatory pathways are intricately linked with oxidative stress and mitochondrial dysregulation. Persistent inflammation leads to excessive reactive oxygen and nitrogen species production, disrupts mitochondrial homeostasis, and hampers cellular energy metabolism, thereby exacerbating neuroinflammatory processes. Consequently, the interrelated roles of inflammation and oxidative stress-driven mitochondrial impairment form a pathogenic feedback loop that enhances vulnerability to depression in RA patients.

### 2.2. Oxidative Stress and Mitochondrial Dysfunction

An increasing corpus of scientific evidence indicates that oxidative stress serves as a pivotal mediator connecting peripheral autoimmune inflammatory processes in RA with central neuroinflammatory mechanisms implicated in depression [20,21]. RA initiates mitochondrial oxidative stress, perturbs mitochondrial bioenergetic equilibrium, and promotes CNS neuroinflammation and bioenergetic dysfunction, thereby contributing to depressive symptomatology [21]. Oxidative stress is a quintessential feature of RA pathology, characterized by excessive generation of ROS that serve as pivotal pro-oxidant and pro-inflammatory mediators [22].

In the inflamed RA synovium, activated macrophages, neutrophils, and fibroblast-like synoviocytes (FLS) produce substantial amounts of ROS via mitochondrial electron transport and NADPH oxidase enzymatic activity [23,24,25]. In MDD, oxidative stress and mitochondrial dysregulation are pivotal pathogenic mechanisms underpinning neuroinflammation and neurodegeneration [26,27]. Within the CNS, activated microglia and astrocytes generate ROS and reactive nitrogen species (RNS) through NADPH oxidase 2 (NOX2) and inducible nitric oxide synthase (iNOS), inducing lipid peroxidation of cellular membranes and oxidative modification of proteins [28,29]. Oxidative stress is intricately associated with neurotransmitter homeostasis disruption. Specifically, it downregulates brain-derived neurotrophic factor (BDNF) expression and hampers synaptic plasticity within the hippocampus and prefrontal cortex, thereby heightening susceptibility to depressive disorders [30]. Research indicates that tilapia skin-derived peptides mitigate oxidative stress and depressive-like behaviors in murine models via modulation of the BDNF/TrkB/CREB signaling cascade [31]. Zhu et al. further elucidated that in C57BL/6 murine models displaying anxiety- and depression-like phenotypes, there is a significant increase in oxidative stress within the hippocampal and hypothalamic regions, concomitant with decreased serotonergic neurotransmitter levels. Restoration of redox homeostasis and 5-HT signaling was achieved through supplementation with tryptophan (Trp) oligopeptides [32]. Dysregulation of 5-HT, noradrenergic, dopaminergic, and glutamatergic neurotransmission—hallmark features of MDD—may be further intensified by oxidative neurotoxicity [32,33,34,35].

Excessive immune cell activation in RA leads to persistent secretion of pro-inflammatory cytokines. Due to the high vulnerability of mitochondria to oxidative stress, somatic mitochondrial DNA (mtDNA) damage readily ensues under these conditions [36]. RA patients demonstrate impaired mitochondrial biogenesis, dysregulated mitochondrial dynamics, and decreased mitochondrial membrane potential, collectively compromising neuronal oxidative phosphorylation, impairing synaptic plasticity, and destabilizing neurotransmitter homeostasis, thereby contributing to the manifestation of depressive symptoms [37,38]. Furthermore, persistent neuroinflammation promotes the generation of ROS within the CNS, intensifying mtDNA oxidative damage and perpetuating microglial activation. These interconnected processes establish a pathogenic feedback loop, ultimately resulting in neuronal impairment and neuropsychiatric disturbances [39,40]. RA-associated depression is intricately connected to dysfunctional mitophagy. Inhibition of the PINK1/Parkin pathway impairs mitochondrial quality control, resulting in the buildup of dysfunctional mitochondria that promote ROS accumulation, mtDNA release, and sustained activation of the NLRP3 inflammasome, thereby exacerbating neuroinflammatory processes [41,42,43]. Decreased mitophagy and suppressed Parkin gene expression have been documented in RA patients, contributing to sustained peripheral immune activation and exacerbated CNS inflammation through the BBB, thereby intensifying neuropsychiatric manifestations [44,45].

In conclusion, oxidative stress and mitochondrial impairment are pivotal in the comorbid pathogenesis of RA and depression, primarily through the escalation of inflammatory pathways and disruption of CNS homeostasis. This disequilibrium is not solely attributable to inflammatory processes; dysbiosis of the gut microbiota further compromises intestinal barrier integrity, facilitates translocation of microbial metabolites into systemic circulation, and intensifies systemic inflammation, thereby aggravating oxidative stress and neuroinflammatory mechanisms. Consequently, oxidative stress, mitochondrial dysfunction, and dysbiosis of gut microbiota collectively serve as critical peripheral mediators in RA-associated depression pathophysiology.

### 2.3. Gut Microbiota Dysbiosis

Gut microbiota dysbiosis serves as a key peripheral contributor to the pathophysiology of depression associated with RA. The persistent inflammatory response intrinsic to RA, alongside prolonged use of DMARDs, can disrupt intestinal microbial homeostasis. Microbial-derived metabolites such as lipopolysaccharides (LPS), short-chain fatty acids (SCFAs), and indole compounds can traverse the BBB and influence CNS functioning through multiple mechanisms, thereby heightening the risk of depression [46].

Patients with RA frequently demonstrate decreased microbiome diversity, proliferation of pathogenic microorganisms, and depletion of advantageous commensal bacteria [47]. Such dysbiosis compromises the integrity of the intestinal barrier, promotes chronic inflammation, and can trigger autoimmune responses [48]. For instance, the proliferation of Collinsella has been associated with the downregulation of tight junction proteins, increased intestinal permeability, and exacerbation of arthritic pathology [49]. Zonulin, a pivotal regulator of epithelial barrier integrity, is secreted by intestinal epithelial cells in response to microbial stimuli, promoting disassembly of tight junctions, enhancing paracellular permeability, and facilitating translocation of microbial metabolites into systemic circulation, thereby initiating systemic inflammatory responses [50]. Elevated serum zonulin concentrations have been observed in RA patients [51]. Similarly, individuals with depression exhibit increased serum levels of zonulin and LPS, implying that bacterial metabolites penetrate a compromised intestinal barrier and contribute to neuropsychiatric manifestations [52]. SCFAs, such as acetate, propionate, and butyrate, produced via microbial fermentation of dietary fibers, are crucial for maintaining intestinal epithelial integrity [53]. RA patients often demonstrate diminished SCFA levels and a reduction in SCFA-producing taxa within the phylum Firmicutes, with exogenous SCFA supplementation exerting protective effects in experimental models of arthritis [54]. In MDD, alterations in gut microbiota composition and metabolic function result in dysbiosis, epithelial barrier dysfunction, and increased intestinal permeability [55]. MDD is characterized by an elevated Bacteroidetes/Firmicutes ratio, with an enrichment of *Bacteroides* spp. and depletion of taxa such as *Blautia*, *Faecalibacterium*, and *Eubacterium* [56,57]. Consequently, microbial metabolites, structural components, and entire microbes may translocate via the compromised intestinal barrier into systemic circulation, potentially amplifying systemic inflammatory cascades [58]. Gut microbiota also modulate the synthesis of neurotransmitters and neuroactive compounds, including gamma-aminobutyric acid (GABA), 5-HT, and dopamine (DA) [59,60]. Dysbiosis can disrupt Trp metabolism, the precursor of 5-HT, thereby impairing neurochemical homeostasis within the CNS [61,62]. SCFAs support epithelial barrier function, enhance tight junction protein expression, and preserve mucosal homeostasis; however, levels are frequently markedly reduced in depressive states. Such depletion exacerbates intestinal permeability and attenuates the negative feedback regulation of the HPA axis, leading to HPA axis hyperactivation and associated pathophysiology [63,64,65].

Gut microbiota dysbiosis not only compromises the local intestinal mucosal barrier but also promotes systemic inflammation by reshaping host immune responses [66]. The gut microbiome can interact with innate immune pattern-recognition receptors, such as Toll-like receptors (TLRs), thereby participating in the pathogenesis of RA [67]. Activation of TLR4 triggers NF-κB signaling, which in turn upregulates IL-6, TNF-α, and COX-2 expression and contributes to RA immunopathology [68,69]. In addition, TLR engagement can drive synoviocyte proliferation and joint destruction through activation of MAPK pathways [70].

## 3. NPs in RA-Associated Depression

Over the previous decade, nanomaterials have demonstrated significant advancements in the pharmacological management of neuropsychiatric and immune-mediated disorders. These nanostructures have been shown to substantially enhance the bioavailability and therapeutic efficacy of various antidepressant and immunomodulatory agents [71,72]. The pathophysiology of RA-associated depression involves complex, interconnected pathways, including persistent inflammatory responses, cytokine overexpression, oxidative stress elevation, dysregulation of the HPA axis, and neurotransmitter imbalances. Nanotechnology-based delivery systems can target these pathogenic mechanisms by modulating inflammatory mediators, regulating oxidative stress, promoting neuroplasticity, and optimizing drug distribution within the CNS. Furthermore, nanoparticle carriers act as efficient vectors, increasing the localized potency of anti-inflammatory drugs, antidepressants, and immunomodulators at affected sites. Current research explores a variety of nano-platforms, including metal-based NPs, liposomes, exosomes, polymeric nanocarriers, quantum dots, and carbon-based nanomaterials. Their roles in RA and the depression are presented in Table 1 and Table 2, respectively, with a focus on their mechanistic relevance, therapeutic advantages, and administration routes. Considering that the etiology of depression associated with RA encompasses multiple pathogenic mechanisms—including sustained inflammatory signaling, oxidative stress elevation, mitochondrial impairment, and neuroimmune dysregulation—the aforementioned key pathways have been comprehensively examined in Section 2. The therapeutic efficacy of the nanocarrier systems discussed herein is contingent upon their capacity to modulate these established pathogenic processes. Consequently, subsequent sections will primarily address differences in targeting specificity, delivery strategies, and immunopathological regulatory outcomes, rather than reiterating the molecular inflammatory cascades.

### 3.1. Organic NPs in RA-Associated Depression

Within the intricate pathophysiological landscape of RA-associated depression, organic NPs afford multiple advantages, such as superior biocompatibility, customizable architectures, and elevated drug-loading efficiencies—facilitating concurrent modulation of peripheral joint inflammation and central neuroinflammatory processes. Surface functionalization further augments their capacity to traverse the BBB, rendering them particularly suitable for the targeted delivery of anti-inflammatory, antioxidative, or neuroprotective therapeutics. Consequently, organic NPs are uniquely positioned to enable dual-site (articular-brain) therapeutic interventions in RA-associated neuropsychiatric conditions.

#### 3.1.1. Liposomal Platforms

Liposomal drug delivery platforms constitute a highly adaptable therapeutic modality applicable across a diverse array of disease states. These carriers are typically composed of natural lipids, synthetic lipids, and surfactants. Rational modulation of their compositional parameters allows for engineering liposomes with pH-sensitive drug release profiles, extended systemic circulation, and other desirable pharmacokinetic properties—forming a structural basis for precise, stable, and effective targeted delivery [92].

RA exhibits hallmark features including synovial inflammation, elevated vascular permeability, and aberrant macrophage activation, collectively creating a pathological microenvironment that facilitates liposomal accumulation. Moreover, the enhanced permeability and retention (EPR) effect observed at inflamed sites promotes passive targeting of liposomes to synovial tissues [71,93]. The inflammatory cascade in RA is driven largely by cytokines such as TNF-α and IL-1β [94]. To enhance localized drug delivery, Jia et al. developed a liposomal dexamethasone (DEX) formulation. This nanocarrier system facilitates targeted DEX accumulation within the synovium, effectively reducing serum concentrations of pro-inflammatory cytokines TNF-α and IL-1β, thereby mitigating joint edema and cartilage destruction [73]. Recent research indicates that liposomal nanocarriers enable prolonged systemic circulation and co-delivery of chemotherapeutic agent doxorubicin (DOX) and phytochemical CUR, presenting a promising strategy for combinatorial therapeutics in RA and neuroinflammatory conditions [95].

Beyond traditional immunosuppressants, liposomal encapsulation serves as a vector for natural bioactive compounds with antioxidant and anti-inflammatory properties aimed at mitigating RA-associated neuroinflammation. Sun et al. demonstrated that liposomal encapsulation of dimethyl CUR (Lipo-DiMC) reduced peripheral leukocyte counts and downregulated expression of dipeptidyl peptidase IV (DPPI) and matrix metalloproteinases (MMP-2/9), thereby alleviating joint swelling and disease severity in collagen-induced arthritis (CIA) rat models [72]. CUR also inhibits pro-inflammatory cytokine-induced NF-κB activation and COX-2 expression, contributing to its anti-inflammatory efficacy. In the CNS, CUR also exerts the aforementioned effects, while further enhancing its antidepressant-like actions [82]. Additionally, it can alleviate neuroinflammation, mitigate oxidative stress, and promote neuroplasticity [96,97]. Triptolide exerts anti-inflammatory actions at both peripheral and central sites; Guo et al. developed folate-conjugated triptolide-loaded liposomes (FA-TP-Lips), which effectively inhibited pro-inflammatory cytokines and osteoclastogenesis [98]. Additionally, triptolide mitigates hippocampal neuroinflammation, modulates microglial polarization, and inhibits p38 MAPK signaling, reversing cytokine imbalances and alleviating depression-like behaviors associated with RA [90]. Capini et al. formulated liposomes co-encapsulating quercetin and methylated bovine serum albumin (BSA) antigens, targeting antigen-presenting cells to suppress NF-κB activation and thus ameliorate joint inflammation and damage in antigen-induced arthritis models [99]. Despite its potent antioxidative and anti-inflammatory profile, quercetin’s poor brain bioavailability underscores the utility of liposomal delivery systems, which enhance stability and direct targeting to neural tissues. Priprem et al. synthesized quercetin-loaded liposomes (~200 nm), administered intranasally, demonstrating BBB bypass via olfactory and cerebrospinal fluid pathways, and improved neurobehavioral outcomes at low doses [100]. Further surface modifications such as PEGylation and nanoemulsion strategies have optimized quercetin brain delivery, with PEGylated liposomes reducing neurotoxicity markers (MDA, 5-HIAA) and increasing antioxidant enzyme activities (GPx, SOD), thereby mitigating depression-like behaviors and supporting their application in neuroinflammatory disorders associated with depression [101].

For stimuli-responsive delivery systems, liposomes can leverage elevated ROS levels characteristic of RA lesions to facilitate on-demand therapeutics release. Song et al. engineered Dex@FA-ROS-Lips utilizing ROS-sensitive lipid components containing thioether linkages. Under heightened ROS conditions, the oxidation of thioethers to sulfoxides or sulfones destabilizes the liposomal bilayer, promoting DEX release, while folate conjugation enables receptor-mediated targeting of folate receptor beta (FR-β) [75]. However, the oxidation of thioether bonds necessitates a threshold ROS concentration; in early RA stages or during remission, subthreshold ROS levels may result in incomplete liposomal disassembly and inadequate drug release, potentially limiting therapeutic efficacy. To mitigate this limitation, Chen et al. encapsulated catalase (CAT) within liposomes, exploiting its enzymatic ability to decompose hydrogen peroxide (H_2_O_2_) into oxygen and water, thereby generating gas that disrupts liposomal integrity and facilitating rapid MTX release while simultaneously scavenging local ROS, thus eliminating reliance on a specific ROS threshold [76]. Nonetheless, conventional liposomes predominantly depend on passive targeting mechanisms; within complex inflammatory microenvironments, they often demonstrate suboptimal drug delivery efficiency and limited retention at disease sites. To address this challenge, Wang et al. designed albumin-modified liposomes that selectively bind to albumin receptors overexpressed in inflamed tissues, thereby enhancing active targeting capabilities. This strategy exhibited higher accumulation and prolonged retention within inflamed tissues, underscoring that active surface modifications can significantly bolster the therapeutic index and safety profile of liposomal nanocarriers [102].

Compared with polymeric and inorganic nanocarriers, liposomes exhibit superior biocompatibility and drug-loading flexibility, making them particularly suitable for anti-inflammatory and antidepressant co-delivery. The increased vascular permeability observed at RA lesions promotes passive accumulation of liposomes within synovial tissue, while ligand-based surface modifications—such as folate and albumin conjugation—enhance active targeting to macrophages or inflamed endothelium. In the context of depression, liposomes can be functionalized with low-molecular-weight moieties to facilitate BBB translocation and deliver anti-inflammatory and antioxidant compounds to CNS targets. However, their limited structural stability and rapid clearance may restrict long-term therapeutic efficacy. Therefore, liposome-based systems are more appropriate for short-term or acute intervention rather than sustained dual-site modulation.

#### 3.1.2. Polymer Nanoparticle Platforms

Polymer NPs fabricated from biodegradable macromolecular materials constitute a versatile class of nano-scale drug delivery vectors distinguished by customizable structural properties, high drug encapsulation efficiency, and tunable pharmacokinetics [103]. Relative to liposomal systems, polymeric nanocarriers provide enhanced regulatory control over drug stability, release profiles, and multifunctional therapeutic payloads [104]. Consequently, they have garnered significant research focus in the targeted treatment of RA and neuropsychiatric conditions [105,106].

Polymeric nanocarriers facilitate both passive and active targeting mechanisms, enabling localized drug accumulation at inflamed synovial tissues while minimizing systemic toxicity through controlled release kinetics. Due to their customizable size parameters, surface functionalities, and copolymer compositions, polymer NPs demonstrate distinct advantages in traversing the BBB and delivering therapeutics to the CNS. Their highly tunable architectures and robust stability profiles make them suitable for sustained-release formulations, dual-drug loading, and multi-barrier penetration strategies. In RA, targeted delivery predominantly focuses on the abatement of hyperactivated macrophages, which exploit receptors such as folate receptor (FR) and scavenger receptor (SR). Conversely, in depression models, strategies often involve surface modifications with polysorbate-80 (PS-80), transferrin (Tf), or biomimetic membrane coatings to enhance cerebral tropism. Exploiting the high expression of FRs on activated macrophages, Tan et al. engineered folic acid (FA)-conjugated poly(lactic-co-glycolic acid)-polyethylene glycol (PLGA-PEG) NPs (FA-NPs/GER), which significantly increased retention at inflamed sites via FR-mediated endocytosis [107]. Similarly, Yusuf et al. developed PS-80-modified PLGA NPs loaded with CUR (PS-80-CUR-NP), mimicking low-density lipoproteins (LDL) to facilitate BBB transcytosis via LDL receptor pathways, while enabling sustained CUR release to improve bioavailability. Behavioral and biochemical analyses confirmed the efficacy of PS-80-CUR-NP in reversing stress-induced depressive phenotypes in murine models [108].

Cholesterol-chitosan conjugates, due to their enhanced biocompatibility and versatile modification capacity, represent promising carriers for concomitant RA and depression therapeutics. Kottarath et al. targeted FR-β overexpression at inflamed sites by constructing anti-FR-β antibody-conjugated cholesterol-chitosan NPs for MTX delivery, underscoring the potential of FR-β-mediated targeting in RA [109]. Fahmy et al. utilized PS-80-modified CUR-chitosan NPs to amplify antidepressant efficacy, attributed to elevated monoamine neurotransmitters and mitigated oxidative stress [110]. Further, He et al. engineered Tf-modified carboxymethyl chitosan NPs (NOCMS-CS-Tf NPs), which demonstrated enhanced brain accumulation in depressive murine models. Collectively, these findings accentuate the versatility and therapeutic promise of chitosan-based nanocarriers in managing RA-associated depression [111]. Dextran, a naturally occurring polysaccharide characterized by low toxicity, biodegradability, and minimal immunogenicity, serves as an effective platform for drug delivery. Activated macrophages overexpress SR and produce high levels of cytokines. Han et al. exploited SR overexpression by utilizing dextran sulfate (DS) as a targeting ligand conjugated to DEX via acid-labile maleic anhydride linkers, forming self-assembled micellar nanostructures (DEX@DS-cad-DEX). These micelles accumulate passively through extravasation through leaky vasculature and inflammatory sequestration (ELVIS) and actively target macrophages via DS-SR interactions, enabling pH-responsive drug release in acidic inflammatory microenvironments [112].

To develop a dual-therapeutic platform addressing both cancer and depression, Shoaib et al. adapted the fluoxetine-dextran nanoparticle (FLX-DEX NP) system, wherein FLX covalently conjugated to aldehyde-functionalized dextran was PEGylated to enhance stability [91]. In vivo studies revealed significant elevations in hippocampal, cortical, and striatal dopamine and 5-HT levels, along with reductions in their respective metabolites DOPAC and 5-HIAA, indicative of targeted CNS delivery rather than nonspecific distribution [113]. This approach offers a conceptual framework for nanotherapeutic intervention in comorbid RA-associated depression. Chen et al. conjugated dimethylamino groups to polydopamine (PDA) NPs, endowing the latter with a positive surface charge. This modification facilitates the sequestration of cell-free DNA (cfDNA) in the joint cavity, thereby reducing proinflammatory factors [114]. PDA NPs also cross the BBB efficiently, restore synaptic integrity markers such as vGLUT1 and PSD-95 in hippocampal and prefrontal regions, ultimately alleviating anxiety- and depression-like behaviors [115]. In LPS-induced inflammatory depression model, Jiang et al. developed microglia-membrane-coated PDA NPs loaded with memantine (Mem) (PDA-Mem@M) to combine BBB penetration with inflammation-targeted therapy, ROS scavenging, and neuroprotection. Mem was released via acid-sensitive Schiff-base linkages, enhancing drug bioavailability at lesion sites [89]. Additionally, CUR’s pH-triggered release via Schiff-base chemistry suggests that pH-responsive NPs offer a strategic modality for targeted therapy in RA-associated depression [116].

In contrast to liposomes, polymeric NPs offer improved structural stability and controllable drug release, which is advantageous for sustained anti-inflammatory and neuroprotective effects. Nevertheless, their relatively complex synthesis and potential polymer-associated toxicity raise concerns for long-term clinical translation. Thus, polymeric systems may be more suitable for chronic RA-associated depression management where prolonged therapeutic exposure is required.

#### 3.1.3. Exosome Platforms

Exosomes, an important subclass of extracellular vesicles (EVs), are naturally secreted via the exocytotic pathway and typically range from 30 to 200 nm in diameter. These nanoscale vesicles contain diverse bioactive cargo-including proteins, lipids, mRNAs, and miRNAs [117]. For instance, macrophage-derived exosomes preferentially accumulate in inflamed tissues, whereas those originating from neurons or glial cells readily cross the BBB and home to regions of neuroinflammation or oxidative stress [118,119]. Thus, in RA-associated depression, exosomes and biomimetic exosome-inspired systems offer an ideal platform for enabling bidirectional drug delivery between peripheral and central compartments.

As natural vectors of miRNAs, exosomes efficiently deliver target-specific miRNAs to recipient cells with minimal immune activation. Moreover, their intrinsic ability to cross the BBB allows exosomes to regulate neuroinflammatory processes and promote neural repair within the CNS, opening novel avenues for depression intervention [120]. Bone marrow-derived mesenchymal stem cell (BMSC) exosomes, for instance, act as vectors of bioactive molecules to modulate RA pathogenesis. Studies indicate that BMSC exosomes significantly suppress pro-inflammatory cytokine in RA rat models [77]. In depression models, chronically elevated corticosterone levels resulting from HPA axis hyperactivity markedly reduce hippocampal miR-26a expression [121]. The downregulation of miR-26a exacerbates hippocampal neuronal apoptosis, oxidative stress, and neuroinflammation, culminating in depressive-like behaviors [122]. Guo et al. demonstrated that BMSC-derived exosomes loaded with miR-26a, administered intravenously to the hippocampus, efficiently restored miR-26a levels [87]. Additional research indicates that BMSC exosomes confer anti-inflammatory, antioxidant, and neurogenic benefits in LPS-induced depression models [123]. In RA, mesenchymal stem cell-derived EVs (MSC-EVs), pretreated with interferon-beta (IFN-β), effectively suppress multiple RA-associated cytokines. Furthermore, MSC-EVs inhibit FLS migration and downregulate surface markers such as CD34 and HLA-DR on FLS populations [124].

To address the imbalance between pro-inflammatory M1 and anti-inflammatory M2 macrophages in RA, Li et al. proposed the use of M2 macrophage-derived exosomes (M2 Exo) as endogenous carriers to co-deliver interleukin-10 (IL-10) plasmid DNA (IL-10 pDNA) and betamethasone sodium phosphate, thereby alleviating arthritis symptoms [125]. Microglia and macrophages, integral components of the mononuclear phagocyte system, also produce exosomes that influence depression-related pathways. Xie et al. found that exosomal let-7e-5p from microglia impairs hippocampal neurogenesis by suppressing Wnt1/β-catenin signaling, an effect reversed by quercetin-mediated downregulation of let-7e-5p, ultimately improving depressive-like behaviors in mice subjected to chronic unpredictable mild stress (CUMS) [86]. These findings further highlight the regulatory capacity of exosomal miRNAs in depression therapy. Neural stem cell-derived exosomes (NSC-EVs) contain multiple neuroregulatory miRNAs involved in intercellular communication. For example, miR-16-5p plays a role in adult hippocampal neurogenesis and the antidepressant efficacy of SSRIs. Conversely, MYB family proteins are associated with neuronal apoptosis. Min et al. reported that NSC-EVs deliver miR-16-5p to suppress MYB expression, thereby attenuating corticosterone-induced neuronal apoptosis and mitigating neuronal injury and behavioral deficits in depressive rat models. This strategy underscores the therapeutic potential of miR-16-5p and NSC-EVs in depression treatment [83]. Addressing gut dysbiosis, impaired serotonin (5-HT) synthesis, and reduced BDNF expression characteristic of depression, Hong et al. developed a plant-derived EV-based (PDEV) intervention. Exosomes isolated from Lepidium meyenii (Maca-EVs) exhibited antidepressant effects in mice subjected to CUMS. Subsequent activation of TrkB and p-AKT signaling mitigated hippocampal and cortical neuronal damage, reversing depressive-like behaviors [126].

As naturally occurring nanoscale vesicles, exosomes exhibit intrinsic targeting properties dictated by their membrane proteins: macrophage-derived exosomes preferentially accumulate within RA-inflamed joints, while neurogenic exosomes effectively penetrate the BBB to access neuroanatomical regions such as the hippocampus and cortex. Despite their advantages, exosome-based systems face challenges in standardization, scalability, and regulatory approval, which currently limit their clinical readiness compared with synthetic nanocarriers.

### 3.2. Inorganic NPs in RA-Associated Depression

Inorganic nanocarriers, as an emerging class of nanoplatforms utilized for targeted drug delivery, demonstrate significant potential in enhancing therapeutic efficacy through selective targeting, sustained-release profiles, and improved solubility and bioavailability of poorly soluble pharmacological agents, thereby optimizing clinical outcomes. The primary categories of inorganic NPs include metallic NPs (e.g., gold, silver, platinum, nickel), metal oxide NPs (e.g., titanium dioxide, manganese dioxide, magnetite, alumina), and semiconductor NPs (e.g., silicon, ceramic materials) [9].

#### 3.2.1. Metallic NPs

RA, a chronic autoimmune inflammatory disorder, involves key pathophysiological interactions between pro-inflammatory M1 macrophages and activated FLSs, resulting in synovial hyperplasia and cartilage degradation. Utilizing glutathione (GSH) as a stabilizing ligand, Chen et al. synthesized gold nanoclusters (Au_25_@GSH), which effectively inhibit NF-κB signaling pathways, induce macrophage polarization from the M1 to the M2 anti-inflammatory phenotype, and decrease expression of cytokines. Additionally, Au_25_@GSH nanoclusters selectively inhibit thioredoxin reductase (TrxR) activity, disturb redox homeostasis in FLSs, and trigger apoptosis via the STAT3-cyclin D1-Bcl-2 axis, thereby mitigating FLS proliferation, migration, and invasion, supporting a safe and efficacious RA therapy [81]. Another approach employed triamcinolone-loaded gold NPs (Triam-AuNPs), which suppress FLS secretion of TNF-α and IL-1β, while simultaneously upregulating anti-inflammatory cytokines such as IL-4 and IL-10, providing bidirectional modulation of FLS-mediated inflammatory processes [127]. Regarding the pathological mechanism of intestinal flora imbalance in RA, Yang et al. employed gold nanospheres (GNS) to achieve targeted regulation of intestinal flora and tryptophan (Trp) metabolism. Specifically, GNS can enrich probiotics and upregulate tight junction proteins, thereby contributing to the repair of the intestinal barrier [128].

Targeting M1 macrophages for phenotypic modulation, folic acid-modified silver NPs (FA-AgNPs) have been employed to induce apoptosis and promote polarization towards the M2 phenotype, resulting in effective RA intervention [129]. Biogenic selenium NPs (SeNPs), characterized by low toxicity, high bioavailability, and potent antioxidant and anti-inflammatory activities, are promising candidates for RA-associated depression management, alleviating oxidative stress, inflammation, and adverse effects linked to conventional therapies [130]. Wang et al. further refined SeNPs with sulfated Ganoderma polysaccharides (SeNPs-SPS), demonstrating dose-dependent downregulation of iNOS, TNF-α, IL-1β mRNA, and upregulation of IL-10 mRNA [74]. Recent mechanistic research has delineated the role of aberrant microglial activation-mediated neuroinflammation and hyperactivation of the JAK2-STAT3 pathway as critical regulators in the progression from stress-induced neural injury to depressive-like behaviors [131]. In a fluoride exposure model, characterized by BBB crossing, neuronal atrophy, monoamine neurotransmitter deficiency, microglial overactivation, and JAK2-STAT3 hyperactivation, SeNPs were used as an intervention. Low-dose SeNPs inhibited nuclear translocation of p-STAT3, reduced IL-1β secretion, reversed fluoride-induced microglial morphological abnormalities, restored cortical dopamine (DA) and norepinephrine (NE) levels, increased neuronal survival, and ameliorated vacuolar degeneration; these neuroprotective outcomes significantly reduced immobility in tail suspension and forced swim assessments, indicating mitigation of fluoride-induced depressive-like behaviors [85].

#### 3.2.2. Metal Oxide NPs

To circumvent hypoxia within the RA synovium, Kim et al. developed mesoporous silica NPs co-modified with manganese ferrite and ceria (MFC-MSNs). This nanoplatform combines oxygen-generating capabilities with ROS scavenging, facilitating macrophage phenotype transition from M1 to M2 and serving as a carrier for MTX for sustained drug release. In CIA models, MFC-MSNs increased synovial oxygen saturation, markedly suppressed HIF-1α expression, reduced M1 macrophage infiltration, and enhanced M2 macrophage proportions, establishing their utility as a novel nanotherapeutic system for RA management via macrophage phenotype modulation [78]. Addressing ROS accumulation and microglial cytokine release in the CNS, Zavvari et al. engineered cerium oxide NPs (CeO_2_ NPs), capable of reversible cycling between Ce^3+^ and Ce^4+^ states, mimicking superoxide dismutase (SOD) and catalase (CAT) activities for continuous ROS scavenging. These nanomaterials lowered hippocampal malondialdehyde (MDA) levels and inhibited microglial activation in ultra-stress models. Furthermore, intracerebroventricular or intrahippocampal administration of CeO_2_ NPs enhanced neurogenesis, upregulated synaptic plasticity markers such as GAP-43, and mitigated neuronal loss in the CA3 hippocampal region [84].

Magnetic iron oxide NPs (IONPs) possess significant potential for integrated diagnostic and therapeutic applications in RA. IONPs are characterized by low cytotoxicity, favorable biocompatibility, superparamagnetic properties, and biodegradability, and are currently the only metal oxide nanostructures approved by the US Food and Drug Administration (FDA) as magnetic resonance imaging (MRI) contrast agents. In targeted therapies, IONPs can be directed via external magnetic fields or functionalized with ligands or antibodies such as αvβ3 integrin to enhance specificity. Photothermal therapy utilizing 220 nm IONPs can elevate local temperatures to approximately 51.7 °C, eliciting notable anti-inflammatory effects [132]. In neurodegenerative contexts, iron deficiency impairs the function of tyrosine hydroxylase and Trp hydroxylase, thereby disrupting neurotransmitter biosynthesis. Saeidienik et al. first reported that γ-Fe_2_O_3_ NPs mitigate LPS-induced depressive-like behaviors. Repeated intraperitoneal administration of 1 or 5 mg/kg γ-Fe_2_O_3_ NPs in rodent models markedly decreased brain inflammatory cytokines and alleviated central inflammatory injury. Fe_2_O_3_ nanoparticle-derived iron also functions as a cofactor in dopamine and 5-HT synthesis, aiding in the restoration of monoaminergic neurotransmitter levels [133]. Additionally, CUR-coated IONPs (Cur-IONPs) further enhance these therapeutic effects. In a reserpine-induced depression model, Cur-IONPs not only rectify iron deficiency-related neurotransmitter synthesis deficits but also leverage CUR’s anti-inflammatory and antioxidant capabilities to inhibit monoamine oxidase activity, reduce neurotransmitter degradation, and decrease malondialdehyde (MDA) and nitric oxide (NO) levels, thereby mitigating the combined impact of inflammation and oxidative stress on neurotransmitter regulation [134].

Among metal oxide nanostructures, Xie et al. evaluated zinc oxide NPs (ZnO NPs) in LPS-induced chronic depression mouse model. Intraperitoneal administration of 5.6 mg/kg ZnO NPs significantly reduced immobility in the tail suspension test (TST) and forced swim test (FST). In the Morris water maze, ZnO NP-treated mice exhibited shortened escape latencies and increased time in the target quadrant, indicating amelioration of depression-related spatial learning and memory deficits. In vivo electrophysiological assessments demonstrated that ZnO NPs enhanced long-term potentiation (LTP) within the dentate gyrus (DG) and restored LPS-induced deficits in synaptic transmission, suggesting neuroprotective effects mediated through the promotion of synaptic plasticity [135].

Inorganic NPs, including gold, silver, selenium, iron oxides, zinc oxide, and ceria, combine structural stability with catalytic activity, functioning as “nanozymes” capable of scavenging ROS, modulating macrophage polarization, and enhancing mitochondrial function. These properties underpin their therapeutic potential in mitigating oxidative stress in RA and addressing neuroinflammation in depression.

### 3.3. Bionic NPs

Recent advances in bioinspired nanoplatforms leveraging cellular mimicry, targeted delivery mechanisms, and therapeutic paradigms have demonstrated capabilities surpassing conventional material classifications. Bionic NPs emulate natural cellular membranes and biofunctionalities, substantially improving nanocarrier stability, target specificity, and immunocompatibility within biological environments. Their primary advantage involves preserving native cell membrane proteins, receptors, and adhesion molecules, conferring extended systemic circulation, enhanced BBB penetration, and immune evasion [136,137]. These innovations inform novel nanodelivery strategies pertinent to complex neuroinflammatory and neurodegenerative co-pathologies.

Zhou et al. developed a macrophage membrane camouflage combined with MMP-responsive drug release mechanism, resulting in DSPE-GPLGVRGC-PEG (DGP) NPs with outer-layer coating. These vehicles facilitate macrophage membrane-mediated targeted delivery, enabling precise release of 2-aminoethoxydiphenyl borate (2-APB) at sites of inflammation. Additionally, encapsulating DEX within microglia membrane-coated PDA NPs enables crossing the BBB and targeting microglia involved in neuroinflammation. In stress-induced models, this approach demonstrates superior efficacy in alleviating depressive behaviors [89]. Previous studies have demonstrated that macrophage membrane-coated NPs can effectively cross the BBB and target intracerebral inflammatory lesions for the treatment of CNS inflammatory disorders, thus exhibiting potential therapeutic value for depression subtypes involving neuroinflammation—necessitating further direct validation [138]. Zhang et al. engineered neutrophil membrane-coated NPs that showed significant therapeutic efficacy in both CIA mouse models and human TNF-α transgenic arthritic mice for treatment and prophylaxis of CIA [139]. In post-glioblastoma therapy, Chen et al. developed a neutrophil-mimicking membrane (NM)-coated PLGA-PEG nanoparticle system loaded with DOX, termed NM-PD, for drug delivery [140]. The neutrophil membrane-coated nanocarrier exploits inflammation-induced homing properties to enhance BBB penetration and lesion targeting, demonstrating antioxidant, anti-inflammatory, and neuroprotective effects in models of neuroinflammation-associated neuropathology. These findings suggest a potential therapeutic approach for depression involving neuroinflammation, although direct validation within depression models remains imperative.

Under neuroinflammatory conditions, cell membrane homology and increased BBB permeability synergistically facilitate nanoparticle entry into the CNS, where they target activated microglia. Given the systemic comorbidity of RA-associated depression, biomimetic NPs theoretically offer the unique advantage of concurrently modulating peripheral joint inflammation and central neuroinflammation, potentially disrupting the pathogenic positive feedback loop of “peripheral inflammation-centrally mediated depressive-like changes.” Although direct evidence in depression remains limited, considering the crucial role of neuroinflammation in the onset and progression of depression, this strategy holds promise for precision treatment of neuroinflammation-associated depressive subtypes. Nevertheless, further investigations are required to validate therapeutic efficacy and safety.

### 3.4. Gene and RNA Nanotherapeutics

Gene and RNA-based therapeutics facilitate programmable, precise modulation of inflammatory and neuroplasticity pathways at the transcriptional or translational level, offering a novel intervention dimension for depression associated with RA. This systemic comorbidity is driven by persistent peripheral immune inflammation and propagated via the “joint-brain axis.” Nanocarrier systems substantially enhance nucleic acid bioavailability within inflamed synovial tissue and central neural targets, overcoming intrinsic challenges such as nuclease-mediated degradation, limited tissue biodistribution and cellular internalization, inadequate endosomal escape, and activation of innate immune responses.

In RA, nanoplatform-mediated gene and RNA therapies have demonstrated promising potential for directly modulating key inflammatory mediators and immune cell phenotypes. For instance, lipid nanoparticle-based delivery of TNF-α siRNA has been utilized to downregulate this pivotal pro-inflammatory cytokine [141]. Beyond suppression of inflammatory factors, gene delivery can also augment anti-inflammatory mediators. Studies employing macrophage-targeting strategies to deliver IL-10 plasmid DNA have achieved joint-specific enrichment through macrophage recruitment at inflamed sites, inducing macrophage polarization, thereby modulating the RA microenvironment and providing insights into localized, sustained anti-inflammatory signaling within complex immune networks [142]. Furthermore, microRNAs (miRNAs), as critical post-transcriptional regulators of immune homeostasis, can be systemically modulated via exosomes or biomimetic nanovesicles. For example, exosomes derived from mesenchymal stem cells carrying miR-223 have been shown to downregulate macrophage NLRP3 inflammasome activation, alleviating RA-associated inflammation. This exemplifies the potential of inflammasome targeting to reprogram innate immune responses [77]. Exosome-mediated miRNA delivery also offers promising avenues for addressing depression-related neurodegenerative pathology, where microglia-derived exosomes containing miR-146a-5p regulate neurogenesis linked to depression, providing direct evidence for nucleic acid interventions aimed at restoring neuroplasticity [143]. Importantly, both peripheral synovial macrophages and central microglia amplify inflammatory signals via NLRP3 inflammasomes and NF-κB pathways, connecting these processes to depression-like behaviors, impaired neurogenesis, and synaptic dysfunction [144]. Recent advancements in crossing the BBB with nucleic acid delivery nanocarriers have yielded lipid NPs capable of intravenous administration, achieving high transfection efficiency in neurons and astrocytes across the CNS with favorable biocompatibility. This establishes a safe, effective platform for CNS RNA delivery [145]. Accordingly, the development of “joint-brain dual-targeting” RNA delivery strategies focusing on shared inflammatory nodes, such as NLRP3, holds promise for simultaneous attenuation of peripheral and central neuroinflammation.

Overall, emerging nanomedicine strategies are progressing from singular delivery platforms toward multifunctional, programmable, and systems-level regulatory modalities. For RA-associated depression, future nanotherapeutics should emphasize pathology-driven rational design over mere material complexity. The integration of biomimetic nanotechnologies with nanoRNA therapeutics could enable comprehensive intervention platforms capable of concurrently modulating immune and neuroinflammatory responses and metabolic dysregulation, thereby advancing precision medicine for depression linked to RA.

### 3.5. Other Nanoparticle Platforms

Beyond the aforementioned systems, emerging nanotechnologies such as dendrimers and carbon-based nanomaterials demonstrate unique potential in treating RA-associated depression.

Li et al. developed a multifunctional nanoplatform based on generation-5 poly(amidoamine) (PAMAM) dendrimers to target RA pathology driven by M1 macrophages. The dendrimers were sequentially modified with 1,3-propanesultone and PEGylated folic acid (FA) to achieve M1 macrophage targeting, while PEGylated α-tocopherol succinate (α-TOS) was incorporated to impart antioxidant activity. Gold NPs encapsulated within the dendrimer, along with a TNF-α inhibitor, facilitate targeted delivery to macrophages, effectively scavenging ROS and suppressing both mRNA and protein expression of TNF-α, thereby offering a promising strategy to alleviate RA-associated depression [79].

Carbon dots (CDs), a novel class of photoluminescent carbon nanomaterials, possess advantageous properties including ultrasmall size, facile synthesis, cost-effectiveness, tunable functionality, excellent biocompatibility, and photostability. He et al. engineered PEG-modified, multi-enzyme-mimicking carbon dot nanocomposites loaded with MTX, designated as CDs2-P@M. These CDs exhibit catalase- and superoxide dismutase-like activities, enabling broad-spectrum scavenging of H_2_O_2_, superoxide, and hydroxyl radicals. PEGylation enhances biocompatibility, diminishes systemic toxicity, and facilitates targeted accumulation within inflamed joints in RA models. Upon homing to arthritic tissues, the system releases MTX to suppress pro-inflammatory cytokine production while spontaneously leveraging the intrinsic catalytic properties of CDs to eliminate excess ROS, promote macrophage polarization toward the anti-inflammatory M2 phenotype, and inhibit osteoclast activation. Notably, Jia et al. proposed N-doped carbon-dot nanozymes (CDzymes) as innovative antidepressant agents, exploiting the mechanistic relationship between oxidative stress, gut microbiota dysbiosis, and gut–brain axis dysfunction in depression. CDzymes significantly ameliorated depressive-like behaviors, increased hippocampal levels of 5-HT and GABA, restored gut microbial diversity, and corrected amino acid metabolism disturbances [88]. Liu et al. developed a nano-delivery strategy for constructing EGCG-loaded tea superparticles (TSPs), using tea proteins as carriers to load epigallocatechin gallate (EGCG). This strategy can alleviate anxiety- and depression-like behaviors in mice by repairing the intestinal mucosal barrier and reshaping the balance of intestinal flora. Compared with synthetic nanoparticle platforms, plant-derived proteins exhibit superior biocompatibility, degradability, and functional adaptability [146]. Nanoparticle-based strategies targeting the gut microbiota represent a promising future paradigm for treating RA comorbid with depression, as they enable precise spatiotemporal regulation of microbial composition and metabolites to rebalance immune-inflammatory pathways and the microbiota–gut–brain axis.

These findings collectively indicate that carbon dots with multi-enzyme activity, low toxicity, and water solubility represent an emerging class of nanomaterials capable of modulating the gut–brain–joint axis, offering promising avenues for targeted intervention in RA-associated depression.

## 4. Targeting Ligands for Drug-Delivery Systems in RA-Associated Depression

Nanomaterials can improve drug solubility and bioavailability [147]. When drugs are encapsulated within NPs, they are shielded from enzymatic degradation, resulting in enhanced stability and therapeutic efficiency. Nanomaterials can also alter pharmacokinetic properties, including absorption, distribution, metabolism, and excretion [148]. Nanoparticle-based delivery prolongs circulation time and reduces clearance, thereby improving therapeutic outcomes and minimizing adverse effects [149,150]. Furthermore, surface modification of NPs with specific ligands enables targeted drug delivery to diseased tissues while sparing healthy tissues, enhancing therapeutic precision and reducing side effects. Their small size additionally facilitates penetration across cellular and tissue barriers, increasing local drug concentrations at target sites. Because some delivery vehicles lack inherent targeting capacity, incorporating targeting ligands onto nanoparticle surfaces is essential for improving delivery specificity. According to the receptors they recognize, these ligands can be divided into two major categories: (i) ligands targeting RA synovium and inflammatory microenvironments, and (ii) ligands targeting pathological features associated with depression (Figure 2).

### 4.1. Targeting the RA Synovium and Inflammatory Microenvironment

In RA, activated macrophages, FLS, and neovascular endothelial cells infiltrate the synovium, upregulating receptors such as CD44, FR-β, and SR, providing molecular targets for nanomedicine delivery [112,129]. Ligands such as hyaluronic acid, folic acid, polysaccharides, and joint-homing peptides have been utilized for selective recognition of inflamed synovium and targeted therapeutics. For example, hyaluronic acid-coated MTX-polyethyleneimine NPs facilitate selective uptake by activated macrophages via CD44 and accumulate in inflamed joints in arthritis models [151]. Folic acid-based targeting leverages FR-β overexpression, with nanomedicines employing folate modifications demonstrating specific engagement with RA synovium [152]. Dextran sulfate, a polyanionic polysaccharide, can bind SR-A on inflammatory macrophages, allowing for their selective internalization, as demonstrated by Kim et al.’s dextran sulfate-based self-assembled NPs [153]. Among peptide ligands, Meka et al. utilized the novel joint-homing peptide ART-1 to modify liposomes for targeted delivery of IL-27 to synovial endothelial cells, markedly enhancing anti-arthritic efficacy and improving safety [154]. Another joint-homing peptide, ART-2, guided DEX-loaded liposomes to arthritic joints in rat and mouse models, effectively suppressing disease progression [155]. For FLS, a key pathogenic cell type, the synovial-fibroblast-homing peptide HAP-1 enables nanocarriers to selectively recognize inflammatory FLS and simultaneously modulate their hypermetabolic state and inflammatory activity [156].

Collectively, nanodelivery systems based on hyaluronic acid, folic acid, dextran sulfate, and joint-homing peptides such as ART-1/ART-2 and HAP-1 have demonstrated robust active-targeting capabilities toward the RA synovium and inflammatory microenvironment in animal models, laying a solid foundation for future nanotherapeutic strategies addressing RA-associated depression.

### 4.2. Targeting the BBB

Extensive clinical and translational studies indicate that BBB dysfunction, neuroinflammation, and altered synaptic plasticity are major pathological underpinnings of depression. Thus, the CNS and its pathological microenvironment constitute critical targets for nanodelivery strategies. Compromised tight junctions and increased BBB permeability are strongly associated with inflammation, stress susceptibility, and behavioral phenotypes in MDD, as supported by multiple reviews [157]. On this basis, NPs can be engineered with brain-targeting ligands, such as Tf, Angiopep-2, lactoferrin, and RVG peptides-or designed to exploit depression-related BBB and neuroinflammatory alterations to achieve active or pathology-dependent CNS accumulation. Tf-receptor-targeted NPs can markedly enhance BBB transcytosis while maintaining good safety profiles, providing a feasible route for delivering antidepressants and neuroprotective drugs [158]. He et al. constructed Tf-modified carboxymethyl-chitosan/chitosan NPs (NOCMS-CS-Tf) and demonstrated enhanced brain accumulation and excellent biocompatibility under depression-related BBB impairment, indicating their potential as a generalizable CNS-targeting nanocarrier for depression therapy [111]. An alternative approach is to bypass the BBB via the nose-to-brain pathway. Several antidepressants have been formulated into intranasal nanoformulations; for example, chitosan-functionalized lipid-polymer hybrid NPs encapsulating paroxetine hydrochloride exhibited optimal particle size, mucoadhesiveness, and BBB permeability, supporting their potential as a nasal-to-brain delivery platform for depression treatment [159]. Additionally, systematic reviews on Tf and cell-specific peptides or aptamers targeting neurons, microglia, and astrocytes have highlighted that surface modification of nanovesicles with such ligands can achieve highly precise cell-type-specific delivery, enabling fine regulation of neuroinflammation, neuroplasticity, and circuit activity-features highly relevant to designing CNS-targeted nanotherapies for depression [160,161].

## 5. Feasibility and Challenges of Dual-Targeting Nanocarriers for RA-Associated Depression

RA-associated depression is fundamentally a systemic comorbid condition driven by a shared pathophysiological backdrop involving peripheral immune-inflammatory imbalance, oxidative stress dysregulation, and CNS dysfunction. Its pathological process encompasses the regulation of local joint inflammatory microenvironments and central neural networks. Based on this core characteristic, the development of dual-targeted nanocarrier systems capable of simultaneously modulating arthritic inflammatory responses and central depressive-like pathology presents a theoretically advantageous approach. However, this strategy faces numerous technical and biological challenges during practical development and clinical translation, necessitating a systematic assessment of its feasibility. Details are provided in Table 3.

### 5.1. Challenges in Designing Dual-Targeted Nanocarrier Platforms

The primary obstacle in developing dual-targeted nanopharmaceuticals is the co-encapsulation of anti-rheumatic and antidepressant agents within a single nanocarrier system. These pharmacological classes present significant disparities in physicochemical characteristics, targeting modalities, and release profiles. Agents used in RA, such as MTX and corticosteroids, are predominantly hydrophilic and designed for accumulation at peripheral inflammatory sites, often leveraging high affinity for inflamed microenvironments. Conversely, CNS-targeting antidepressants like FLX and Mem require traversing the BBB, typically exhibiting higher lipophilicity and requiring precise control over release timing. These inherent differences pose substantial challenges to synchronized delivery and compartmentalized release within a unified nanocarrier system. Furthermore, the targeting mechanisms for joint synovium and CNS tissues differ fundamentally. RA lesion targeting primarily depends on the EPR (enhanced permeability and retention) and ELVIS (extravasation through leaky vasculature and subsequent inflammatory cell-mediated sequestration) effects for passive accumulation, whereas CNS penetration necessitates receptor-mediated transcytosis mechanisms or exploitation of pathological BBB permeability alterations. Consequently, the critical design challenge in dual-targeted nanocarriers is to reconcile peripheral inflammatory targeting with efficient CNS delivery within a single platform.

### 5.2. Strategies for Co-Loading and Hierarchical Drug Release

Recent advancements have demonstrated partial feasibility of dual-targeted nanoparticle delivery systems by capitalizing on nanocarrier structural compartmentalization to enable spatial separation and functional coordination of diverse therapeutic agents. For example, Ail et al. engineered a thermosensitive nanoliposome-in situ gel system encapsulating hydrophilic FXT within the aqueous core and embedding lipophilic embelin (EMB) within the phospholipid bilayer, facilitating simultaneous delivery of hydrophilic and hydrophobic drugs within a unified lipid vesicle platform [162]. Future research may involve designing polymeric NPs with hydrophobic cores and hydrophilic shells or hybrid lipid-polymer systems to enhance compatibility, enabling encapsulation of antidepressants within hydrophobic regions and loading anti-inflammatory or immunosuppressive agents in hydrophilic outer layers.

Stimuli-responsive nanocarriers offer additional technological advancements for dual-targeted therapy. The microenvironment within RA lesions is characterized by elevated ROS, acidic pH, and hypoxia, which can be exploited to trigger localized drug release of anti-inflammatory agents. Concurrently, crossing the BBB can be facilitated via mechanisms such as sustained or secondary stimuli-triggered release of antidepressants, allowing for temporally and spatially tiered drug delivery. This hierarchical approach addresses delivery conflicts among target tissues, thereby enhancing overall therapeutic specificity.

### 5.3. Bionic and Membrane-Coated NPs for Dual-Targeting Strategies

The progression of biomimetic nanotechnology introduces innovative methodologies for dual-targeted delivery systems. Employing cell membrane cloaking strategies, such as macrophage and microglia membrane coatings, maintains the integrity of native membrane proteins and adhesion molecules, thereby conferring immune evasion properties and targeted homing capabilities to pathological tissues. Macrophage membrane-encapsulated NPs demonstrate preferential accumulation within inflamed synovial joints, while membrane structures derived from central immune cells facilitate recognition and retention in neuroinflammatory regions. Additionally, exosomes and their biomimetic nanovesicles—characterized by intrinsic BBB penetration and low immunogenicity—are considered optimal vectors for bidirectional modulation within the joint–brain axis. Modifications to exosomal surface proteins or internal nucleic acids are aimed at enabling synchronized regulation of immune-inflammatory signaling pathways and neuroplasticity mechanisms, offering a promising framework for systemic therapeutic interventions in RA-associated depression. RNA-based nanotherapies utilizing macrophage- or neutrophil-derived delivery systems for siRNA, antisense oligonucleotides (ASO), or mRNA target pathways such as TNF-α, IL-6/JAK-STAT, and NLRP3 inflammasome, thereby reducing synovitis and inflammation-associated depressive symptoms. Furthermore, these nanoparticle-based cellular carriers facilitate targeted delivery to modulate microglial inflammatory responses, kynurenine pathway activity, or glutamate excitotoxicity, enabling a synergistic therapeutic approach for RA-associated depression.

### 5.4. Molecular Foundations of Potential Synergistic Therapeutic Outcomes

From a mechanistic standpoint, peripheral inflammatory processes and central neuroinflammation engage in reciprocal neuroimmune crosstalk, forming self-perpetuating feedback amplification loops. Advancements in multi-targeted nanodelivery platforms aim to concurrently intervene across various tissue compartments and pathogenic pathways, not merely through additive effects but via coordinated modulation of critical pathological nodes to optimize systemic therapeutic efficacy. In this context, dual-targeted nanodelivery systems afford the capacity to simultaneously disrupt these feedback circuits rather than rely on the cumulative effects of multiple drugs. Effective suppression of peripheral inflammation can restore the integrity of the BBB, reducing the ongoing infiltration of inflammatory mediators into the CNS [163,164]. Conversely, mitigating neuroinflammation and neurotransmitter imbalances-through modulation of neuroimmune axes—can exert feedback inhibition on peripheral immune activation, thereby attenuating inflammatory signaling across multiple levels. This bidirectional regulatory mechanism facilitates the restoration of HPA axis homeostasis, enhances mitochondrial function, and maintains redox balance, collectively yielding a synergistic amplification of therapeutic outcomes.

## 6. Challenges, Clinical Translation, and Future Perspectives

Despite exhibiting considerable potential in the modulation of pathology associated with RA and depression, the translation of nanomedicine from laboratory research to clinical application continues to encounter multiple barriers. Particularly in the context of RA-associated depression-a complex comorbidity involving dual pathological axes of peripheral immune dysregulation and CNS pathology-nano-scale therapeutic strategies must demonstrate not only superior efficacy but also rigorous long-term safety, immunocompatibility, and manufacturability considerations.

### 6.1. Long-Term Biosafety, Immune Responses, and Limitations

The therapeutic promise of nanomedicine for RA-associated depression primarily rests on extensive preclinical data. However, uncertainties regarding long-term biocompatibility and the extrapolation of existing evidence constitute fundamental obstacles to clinical translation. Current studies indicate that certain inorganic nanomaterials-such as metal oxides or carbon-based nanostructures-may pose risks of slow degradation or tissue accumulation, potentially resulting in organ toxicity and metabolic burden upon chronic exposure. Even organic nanocarriers, which are generally regarded as biocompatible, might elicit hepatic, splenic, or immune system perturbations under repeated dosing regimens. Evidence suggests that some nanomaterials can activate complement pathways or induce nonspecific inflammatory responses, which may accelerate clearance and diminish therapeutic efficacy [165]. Regarding biomimetic nanocarriers and exosome delivery platforms, although their immuno-compatibility tends to be higher, factors such as cellular origin, membrane protein heterogeneity, and manufacturing process stability can significantly influence immunological behaviors and batch-to-batch consistency. In a chronic comorbid setting requiring prolonged intervention-such as RA-associated depression-maintaining therapeutic efficacy while avoiding immune tolerance or aberrant immune activation remains an urgent challenge in nanotherapeutic design.

It is noteworthy that the aforementioned safety uncertainties are often magnified within existing evidentiary frameworks. Most current studies rely on single-disease animal models-for example, models of pure arthritis or depression that are inadequate to recapitulate the complex phenotype of patients experiencing long-term coexistence of peripheral chronic inflammation, neuroimmune imbalance, and behavioral changes. Consequently, the therapeutic effects observed in preclinical nanoplatform studies may be overestimated when extended to real-world clinical scenarios. Additionally, substantial heterogeneity exists across studies regarding model construction, dosing regimens, administration frequency, and endpoints related to behavior and inflammation, severely limiting cross-study comparability. Of particular concern is the prevalence of studies employing acute or short-term administration paradigms, which do not align with the clinical necessity for sustained, repeated therapy inherent in RA-associated depression. This discrepancy further exacerbates the translational uncertainty. Furthermore, systematic validation of long-term safety, immunogenicity, cumulative dosing effects, and BBB targeting stability across species and stages of disease remains lacking. Head-to-head comparisons with free drugs or standard combination therapies are limited, and very few studies simultaneously evaluate both joint inflammatory control and neuropsychiatric outcomes within the same comorbid model, obscuring the true advantage of nanoplatforms within the holistic “joint-brain axis” modulation.

Therefore, future investigations must undergo a paradigm shift at the level of computational modeling and translational validation: transitioning from isolated disease models to validated, standardized RA-depression comorbidity models; expanding from short-term pharmacodynamic assessments to systematic evaluations of long-term safety profiles, immunological compatibility, and biocompatibility under repeated dosing regimens; and incorporating critical parameters such as chronic toxicity, immune responses, and clinical feasibility early within the research pipeline. Only through the iterative refinement of safety datasets and disease-specific models can the therapeutic potential of nanomedicine in RA-associated depression be robustly substantiated and effectively translated into clinical intervention.

### 6.2. Barriers to Clinical Translation

Despite the promising outlook of nanotherapeutics in targeting RA and depression-related comorbidities, substantial translational barriers hinder progress from preclinical validation to routine clinical implementation. These challenges are systemic, encompassing aspects such as disease modeling fidelity, efficacy assessment frameworks, scalable manufacturing processes, and regulatory pathways.

In particular, issues related to the scale-up of nanopharmaceutical manufacturing and quality control are pivotal during clinical translation. Compared to conventional small-molecule therapies, nanocarriers impose more rigorous requirements on parameters including particle size uniformity, surface charge stability, drug loading efficiency, and batch-to-batch reproducibility. For multi-component, biomimetic, or engineered nanostructures, production frequently involves complex physical and biological processes, thereby increasing technical difficulties associated with industrial-scale manufacturing. These complexities can elevate production costs and give rise to inter-batch variability in biodistribution and therapeutic efficacy, which in turn magnifies uncertainties in clinical trial data and translational success. It is noteworthy that the majority of nanomedicine approaches under investigation remain at the preclinical stage, predominantly validated in rodent models, with no nanoplatforms yet achieving regulatory approval specifically for the management of depression comorbid with RA. Currently, regulatory frameworks predominantly adhere to traditional pharmacological evaluation paradigms, lacking standardized or conclusive criteria for multifunctional drug co-delivery systems, biomimetic nanostructures, or gene delivery nanocarriers. These unresolved safety and regulatory issues constitute primary bottlenecks hindering clinical translation. Future research should utilize more clinically relevant animal models exhibiting comorbid conditions, incorporate long-term safety and pharmacokinetic assessments, and evaluate manufacturability and regulatory compliance of scalable nanotherapeutic production early in the development process. Enhancing integrated feedback mechanisms between fundamental research and clinical requirements is vital to bridging translational gaps in nanomedicine for RA-associated depression.

### 6.3. Future Directions and Clinical Prospects

Advancing nanotherapeutic strategies for RA-associated depression necessitates a shift from material-centric design towards mechanism-based rational engineering. Deepening insights into the dynamic interactions within inflammation-neuro-metabolic pathways are critical for developing more targeted, multifunctional nanocarriers. Furthermore, integrating nanomedicine with precision medicine approaches-personalizing interventions based on individual inflammatory and depressive phenotypes-may improve therapeutic efficacy. Interdisciplinary collaboration among material scientists, immunologists, neuroscientists, and clinicians will be essential to expedite bench-to-bedside translation. As safety evaluation protocols for biomaterials are refined and regulatory pathways elucidated, nanomedicine is positioned to assume increasingly central role in the systemic management of the complex comorbidity of RA and depression.

## 7. Conclusions

RA-associated depression is not merely a coexistence of peripheral synovitis and depressive symptoms but represents a multifaceted systemic comorbidity driven by persistent immune-mediated inflammation. This pathological interrelation involves complex mechanisms, including BBB disruption, neuroimmune dysregulation, mitochondrial dysfunction mediated by oxidative stress, and gut microbiota dysbiosis. From an integrated “joint–CNS–gut” axis perspective, this review systematically consolidates critical pathogenic pathways relevant to RA-associated depression and summarizes recent advancements in nanocarrier-based therapeutics, including organic nanomaterials, inorganic nanomaterials, exosomes, and biomimetic nanosystems, designed to modulate peripheral inflammation and central neuroinflammation. Current research suggests that nanodelivery platforms facilitate targeted accumulation in inflamed synovial tissue via EPR/ELVIS effects and possess the capacity to traverse the BBB through ligand functionalization, membrane mimetics, and physicochemical property modulation, establishing a technical basis for concurrent intervention in joint and neuroinflammatory processes. Compared to traditional treatments, nanomedicine’s principal advantage resides in its capacity for integrated regulation across multiple pathogenic domains—immune activation, oxidative stress, and neuroplastic changes—extending beyond mere enhancement of pharmacokinetic exposure. Future nanotherapeutic strategies for RA-associated depression should transition from material-centric development toward rational design centered on disease mechanisms and systemic modulation. Through precise integration of targeted delivery, controlled release, and multifunctional regulation, these approaches aim to disrupt the “joint–brain” pathogenic feedback loop, thereby providing a more translationally feasible therapeutic paradigm for managing this complex comorbidity.

## Figures and Tables

**Figure 1 pharmaceuticals-19-00094-f001:**
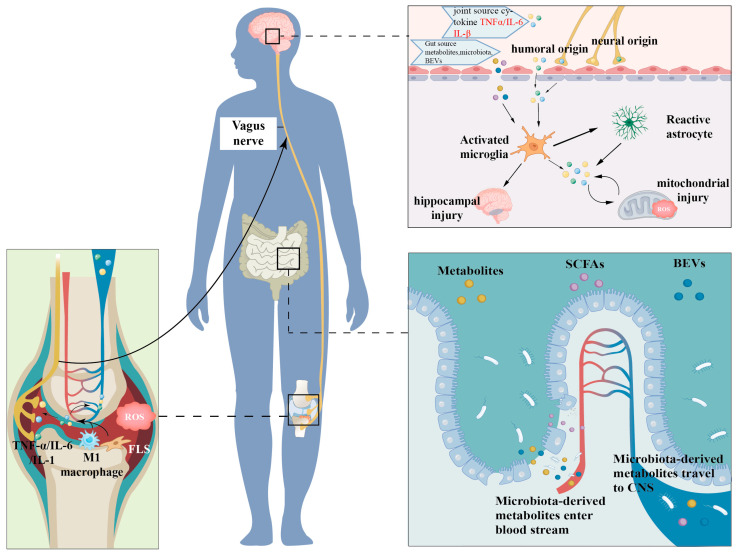
Comorbidity Mechanisms of Rheumatoid Arthritis-Associated Depression.

**Figure 2 pharmaceuticals-19-00094-f002:**
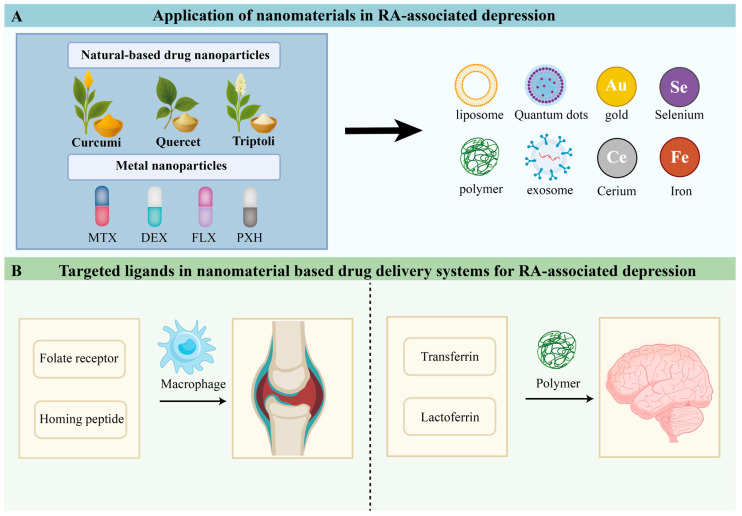
(**A**) A variety of novel nanomaterials have been applied in the treatment of RA-associated depression, including organic NPs and inorganic NPs, which are loaded with natural components or therapeutic agents. (**B**) Targeting ligands for drug-delivery systems in ra-associated depression.

**Table 1 pharmaceuticals-19-00094-t001:** NPs in RA.

No.	Model	Drug	NPs	Administration Route	Pathway	Effect	Reference
1	CIA rats	Dimethylcurcumin (DiMC)	Lipo-DiMC	Intra-articular Injection	DPPI, MMP-2/9	Reduces circulating neutrophils and lymphocytes, suppresses abnormal DPPI and MMP-2/9 expression, and alleviates joint swelling and arthritis severity.	[72]
2	AIA rats	Quercetin + methylated BSA antigen	Quercetin/antigen co-loaded liposomes	Intravenous Injection	NF-κB	Modulates APCs and inhibits NF-κB activation, markedly reducing inflammation and protecting joints.	[73]
3	RAW 264.7 macrophage	-	SeNPs-SPS (selenium NPs with sulfated polysaccharides)	-	NF-κB (IκBα); MAPK (JNK1/2, p38)	Downregulated iNOS, TNF-α, IL-1; upregulated IL-10; inhibited IκBα phosphorylation and JNK/p38 activation, showing strong anti-inflammatory effects.	[74]
4	AIA rats	DEX	Dex@FA-ROS-Lipo (ROS-responsive folate liposomes)	Subcutaneous Injection	ROS-triggered thioether oxidation	High ROS oxidizes thioether lipids to sulfoxides/sulfones, causing liposomal disassembly and DEX release; folate enables FR-β-mediated targeting of inflamed RA tissues.	[75]
5	CIA rats	MTX	Catalase-loaded folate liposomes	Intravenous Injection	H_2_O_2_ decomposition/ROS	Encapsulated catalase decomposes H_2_O_2_ into O_2_, generating pressure that disrupts the liposomal membrane and rapidly releases MTX while scavenging ROS, independent of a ROS threshold.	[76]
6	AIA rats	miR-223	BMSC-derived exosomes	Subcutaneous Injection	NLRP3 inflammasome	miR-223 bound NLRP3 mRNA 3′UTR, downregulated NLRP3, suppressed IL-1β, TNF-α, IL-18, attenuated RA joint inflammation and tissue damage.	[77]
7	AIA rats	MTX	MFC-MSNs (Mn-ferrite/ceria-modified mesoporous silica NPs)	Intra-articular Injection	HIF-1α; ROS; macrophage polarization	Generated O_2_ and scavenged ROS, promoted M1→M2 macrophage shift; reduced HIF-1α, alleviated swelling/hyperthermia, improved motor function and cartilage preservation.	[78]
8	CIA rats	TNF-α inhibitor	Multifunctional PAMAM-dendrimer-entrapped Au NPs	Intra-articular Injection	ROS; TNF-α	PEG-FA targeting + PEG-α-TOS antioxidant units + internal AuNPs + TNF-α inhibitor synergistically scavenged ROS and suppressed TNF-α mRNA/protein expression.	[79]
9	CIA rats	MTX	CDs2-P@M (PEG-modified multienzyme carbon dots with MTX)	Intravenous Injection	Catalase/SOD-like ROS scavenging; M2 polarization	Removed H_2_O_2_, superoxide, and hydroxyl radicals; released MTX to suppress inflammatory cytokines; promoted M2 macrophage polarization and inhibited osteoclast activation.	[80]
10	AIA rats	-	Au_25_@GSH (glutathione-stabilized gold nanoclusters)	Intravenous Injection	NF-κB; TrxR; STAT3-cyclin D1-Bcl-2	Inhibited NF-κB and promoted M1→M2 shift; reduced TNF-α and IL-6; selectively inhibited TrxR in FLS and induced apoptosis via STAT3-cyclin D1-Bcl-2, suppressing FLS proliferation/migration/invasion.	[81]

Abbreviations: AIA, adjuvant-induced arthritis; CIA, collagen-induced arthritis; MMP, matrix metalloproteinases; APCs, antigen-presenting cells; SOD, superoxide dismutase; TrxR, thioredoxin reductase; DEX, dexamethasone; MTX, methotrexate; Lipo, liposomes.

**Table 2 pharmaceuticals-19-00094-t002:** NPs in Depression.

No.	Model	Drug	NPs	Administration Route	Pathway	Effect	Reference
1	LPS-induced depression and anxiety rats	Curcumin	CUR-NLCs (curcumin nanostructured lipid carriers)	Intraperitoneal Injection	NF-κB, COX-2	In LPS-induced depression models, inhibit p-NF-κB, downregulate TNF-α and COX-2, preserve neuronal integrity, and enhance antidepressant and anxiolytic effects.	[82]
2	CORT-induced depression rats	miR-16-5p	Neural stem cell-derived EVs (NSC-EVs)	Tail Vein Injection	miR-16-5p/MYB	Suppressed MYB expression, reduced corticosterone-induced neuronal apoptosis, improved neurodegeneration and depressive-like behaviors.	[83]
3	CUMS-induced depression mice	-	CeO_2_ NPs	Intracerebroventricular Injection	SOD/CAT-mimetic ROS scavenging; IL-6	Eliminated ROS, reduced hippocampal MDA and IL-6, protected neurons, enhanced neurogenesis; stronger antioxidant/anti-inflammatory efficacy than FLX.	[84]
4	NaF-induced depression-like behaviors mice	-	Selenium NPs	Oral Gavage	JAK2-STAT3	Inhibited STAT3 nuclear translocation, reduced IL-1β, restored microglial morphology, recovered DA/NE, enhanced neuronal survival, reduced vacuolar degeneration, improved depression-like behavior.	[85]
5	CUMS-induced depression mice	let-7e-5p cargo	Microglia-derived exosomes	Oral Gavage	Wnt1/β-catenin	let-7e-5p inhibited Wnt1/β-catenin, impaired hippocampal neurogenesis; quercetin downregulated exosomal let-7e-5p and reversed CUMS-induced depressive-like phenotypes.	[86]
6	CORT-induced depression rat	miR-26a	BMSC-derived exosomes carrying miR-26a	Tail Vein Injection	miR-26a/oxidative-stress regulation	Increased hippocampal miR-26a and SOD, reduced MDA and LDH, restored redox balance, suppressed TNF-α and IL-1β, improved depressive-like behaviors.	[87]
7	CUMS-induced depression mice	-	N-doped carbon dot nanozymes (CDzymes)	Oral Gavage	Oxidative stress; gut–brain axis	Increased hippocampal 5-HT and GABA, restored gut microbiota diversity and amino-acid metabolism, alleviated depressive-like behaviors in CUMS rats.	[88]
8	CRS-induced depression mice	Memantine	Microglia-membrane-coated PDA NPs (PDA-Mem@M)	Intravenous Injection	TLR4/NF-κB inhibition; ROS scavenging; microglial M1 to M2 repolarization	Reduces neuroinflammation, protects synapses, improves depressive-like behaviors	[89]
9	CUMS-induced depression mice	Triptolide	Folate-modified triptolide liposomes (FA-TP-Lips)	Intraperitoneal Injection	p38 MAPK-mediated neuroinflammation	Reduces hippocampal inflammation, modulates microglia, alleviates pain-associated depression	[90]
10	HeLa cells	FLX	FLX-dextran NPs	-	Monoamine neurotransmitter regulation	Enhances targeted brain delivery and antidepressant efficacy	[91]

Abbreviations: LPS, Lipopolysaccharide; CUR, Curcumin; CORT, corticosterone; CUMS, chronic unpredictable mild stress; EVs, extracellular vesicles; FA, folic acid; MDA, malondialdehyde; LDH, lactate dehydrogenase; MYB, MYB proto-oncogene; NSC, neural stem cell; FLX, Fluoxetine.

**Table 3 pharmaceuticals-19-00094-t003:** Comparison of single-target and dual-target nanomedicine strategies for RA-associated depression.

No.	Dimension	Single-Target Nanomedicine Strategies	Dual-Target Nanomedicine Strategies
1	Primary therapeutic targets	Either peripheral joint inflammation (synovium, FLS, macrophages) or central depression-related pathology (microglia, neurons)	Concurrent targeting of peripheral arthritic inflammation and central neuroinflammation/neuroplasticity alterations
2	Pathophysiological rationale	Treats RA and depression as relatively independent disease entities	Based on a systemic positive feedback loop linking peripheral inflammation, BBB dysfunction, central neuroinflammation, and depressive-like behaviors
3	Major mechanisms of action	Local suppression of inflammatory mediators in joints or isolated modulation of neurotransmission and neuroinflammation in the CNS	Simultaneous regulation of peripheral immune inflammation, oxidative stress, BBB integrity, central neuroinflammation, and neuroplasticity
4	Nanodelivery dependence	Relies on EPR/ELVIS effects for accumulation in inflamed joints or on BBB-crossing strategies for CNS delivery	Integrates inflammation-homing effects, receptor-mediated BBB transcytosis, and biomimetic membrane targeting to achieve multi-organ distribution
5	Representative nanoplatforms	MTX-loaded liposomes, polymeric NPs delivering anti-inflammatory drugs, brain-targeted nanocarriers carrying antidepressants	Co-delivery nanoplatforms carrying anti-rheumatic and antidepressant agents, exosome-based or biomimetic membrane-coated NPs, multifunctional nanozymes
6	Drug loading and release profiles	Single-site accumulation with relatively simple release kinetics	Hierarchical or stimulus-responsive release: inflammation-triggered drug release in joints followed by sustained or secondary release in the CNS after BBB crossing
7	System-level synergistic effects	Therapeutic effects largely confined to a single organ or pathological level	Breaks the pathological positive feedback loop along the joint–brain axis through bidirectional neuro-immune regulation
8	Key advantages	Simpler design and clearer translational pathways	Better alignment with the complex comorbid pathophysiology of RA-associated depression and greater potential for systemic therapeutic efficacy
9	Major challenges	Limited efficacy in controlling comorbid conditions and higher relapse risk	Increased design complexity, including challenges in drug co-loading, targeting coordination, and long-term biosafety
10	Current research stage	Mostly validated in single-disease animal models	Primarily at early or proof-of-concept stages, with limited validation in comorbid models or clinical settings

## Data Availability

No new data were created or analyzed in this study. Data sharing is not applicable to this article.

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
