# Peer review of "Nanomedicine-Driven Therapeutic Strategies for Rheumatoid Arthritis-Associated Depression: Mechanisms and Pharmacological Progress"

_pharmaceuticals, 2026, doi:10.3390/ph19010094_

Round 1
Reviewer 1 Report
Comments and Suggestions for Authors
The authors present a relevant and timely topic concerning the application of nanomedicine for the management of comorbid depression in rheumatoid arthritis (RA).
The manuscript outlines recent developments in nanoparticle-based delivery strategies, emphasizing inorganic and organic nanomaterials, with the objective of proposing a dual-targeting therapeutic framework.
While the topic holds promise, several key areas require substantial revision and deeper analysis before the manuscript can be considered suitable for publication. The descriptions currently offered are often too concise and lack the depth necessary for a comprehensive review article.
My major comments and required revisions are summarized below:
Major Comments and Required Revisions
- Addressing Key Challenges in Novel Strategies:
- The current manuscript does not sufficiently address the significant practical challenges associated with developing a dual-targeting framework.
- The authors must significantly expand their discussion on the feasibility of simultaneous delivery of anti-RA and antidepressant drugs in a single nanocarrier.
- A deeper analysis is needed regarding the potential synergistic effects and enhanced therapeutic outcomesanticipated from such combination approaches.
- Enhancements in the Introduction Section:
- The introduction should provide a stronger rationale for the need for nanomedicine in this context.
- The authors must explicitly include a section detailing the limitations of conventional therapeutic approaches, specifically addressing issues such as poor bioavailability, systemic toxicity, and critical blood-brain barrier (BBB) impermeability.
- Furthermore, a clear statement defining the scope and unique contribution of this specific reviewis needed to position the work effectively within existing literature.
- Expansion of Advanced and Emerging Directions:
The manuscript would benefit from expanded sections on 'Advanced and Emerging Directions', particularly Biomimetic Nanotechnology and Gene Therapy Approaches. Adding recent developments and future perspectives in these areas would enhance the paper's depth and relevance. These aspects should be substantially expanded to provide a complete understanding of the underlying mechanisms and potential. Descriptions should move beyond surface-level summaries.
- Inclusion of Clinical Translation and Future Perspectives:
- A critical discussion regarding the translational pathway is missing.
- A new, dedicated section must be added before the conclusion addressing "Challenges, Clinical Translation, and Future Perspectives."
- This section should cover practical considerations such as long-term biosafety, toxicity, immune response of nanomaterials, and regulatory hurdles.
Author Response
Thank you for your valuable peer review comment. We have carefully incorporated and implemented the suggested revisions, and have prepared a detailed response outlining the modifications. Please refer to the accompanying document for specific changes.

Reviewer 2 Report
Comments and Suggestions for Authors
The article entitled “Nanomedicine-Driven Therapeutic Strategies for Rheumatoid Arthritis Associated Depression: Mechanisms and Pharmacological Progress” is a review work specifically focused on the Nanomedicine-Driven Therapeutic Strategies for Rheumatoid Arthritis. Authors are encouraged to consider the following comments.
- The manuscript provides an excellent and timely review of nanomedicine for RA-depression comorbidity; integrating the three core mechanisms is a major strength.
- Table 1 is a valuable resource, but a short explanatory caption summarizing its organization and key takeaways would help readers navigate it more intuitively.
- Could you briefly comment on the clinical translation challenges for these nanoplatforms such as safety, scale-up, and regulatory pathways in the conclusion to ground the promising preclinical data?
- The figures are referenced but not included in this draft. Including them will significantly aid visual understanding of the complex pathways discussed.
- Consider adding a short section or paragraph on limitations of current studies, such as the reliance on mono disease animal models, to provide a more balanced perspective.
- The writing is clear overall, but a careful proofread is needed to catch minor grammatical slips and ensure consistent formatting of references and in-text citations.
Author Response

(The authors gave the same response as above.)

Reviewer 3 Report
Comments and Suggestions for Authors
- The manuscript is excessively long and, in several sections, highly repetitive, particularly in the descriptions of inflammatory pathways (like NF-κB, TNF-α, IL-6, oxidative stress) across different nanoparticle platforms. The authors should substantially condense repetitive mechanistic explanations and focus more on comparative insights and conceptual synthesis.
- Although the review is rich in examples, it remains largely descriptive. A more critical perspective is needed. The authors should explicitly discuss the limitations, risks, and unresolved challenges of nanomedicine approaches, including long-term toxicity, immunogenicity, biodistribution variability, manufacturing scalability, and regulatory hurdles.
- The translational relevance is not sufficiently emphasized. Most cited studies are preclinical animal models. The authors should clearly distinguish between preclinical and clinically validated evidence and comment on the feasibility of translating these nanoplatforms into clinical practice.
- The central concept of “dual targeting” (joint–brain or peripheral–central modulation) should be more clearly articulated. A dedicated conceptual framework or summary figure/table comparing dual-targeting strategies versus single-site approaches would significantly strengthen the manuscript.
- The section on gut microbiota and gut–brain–joint axis interactions is scientifically relevant but weakly integrated into the nanomedicine discussion. The authors should clarify how nanocarriers specifically modulate microbiota-related pathways rather than treating this topic as a parallel mechanism.
- Table 1 is informative but very dense. The authors should consider simplifying it or splitting it into separate tables (e.g., RA-focused vs. depression-focused nanotherapies) and clearly indicating administration routes and disease models.
- The conclusion section should be strengthened. It currently summarizes content but does not sufficiently highlight future research directions, unmet needs, or priorities for the field.
- Abbreviations should be checked for consistency and defined at first appearance (ELVIS/EPR, FLS, CUMS, UCMS, etc).
- The balance between organic, inorganic, and exosome-based systems could be improved, as exosomes receive disproportionately detailed coverage compared to other platforms.
Author Response

(The authors gave the same response as above.)

Round 2
Reviewer 1 Report
Comments and Suggestions for Authors
Revised manuscript suitable for publication in Pharmaceuticals
Reviewer 2 Report
Comments and Suggestions for Authors
The overall review paper is good and it creates a sense of interest to reader.